# Channel Prediction Based on BP Neural Network for Backscatter Communication Networks

**DOI:** 10.3390/s20010300

**Published:** 2020-01-05

**Authors:** Jumin Zhao, Hao Tian, Deng-ao Li

**Affiliations:** 1College of Information and Computer, Taiyuan University of Technology, Taiyuan 030024, China; tianhao407@163.com; 2Technology Research Center of Spatial Information Network Engineering of Shanxi, Taiyuan 030024, China; lidengao@tyut.edu.cn; 3College of Data Science, Taiyuan University of Technology, Taiyuan 030024, China

**Keywords:** backscatter communication, channel prediction, link burstiness, acceleration

## Abstract

Backscatter communication networks are receiving a lot of attention thanks to the application of ultra-low power sensors. Because of the large amount of sensor data, increasing network throughput becomes a key issue, so rate adaption based on channel quality is a novel direction. Most existing methods share common drawbacks; that is, spatial and frequency diversity cannot be considered at the same time or channel probe is expensive. In this paper, we propose a channel prediction scheme for backscatter networks. The scheme consists of two parts: the monitoring module, which uses the data of the acceleration sensor to monitor the movement of the node itself, and uses the link burstiness metric β to monitor the burstiness caused by the environmental change, thereby determining that new data of channel quality are needed. The prediction module predicts the channel quality at the next moment using a prediction algorithm based on BP (back propagation) neural network. We implemented the scheme on readers. The experimental results show that the accuracy of channel prediction is high and the network goodput is improved.

## 1. Introduction

The core mission of the Internet of Things (IoT) is ubiquitous data-awareness, wireless-based data transmission, and intelligent data processing. With the development of 5G, cloud computing, big data, and artificial intelligence technologies, a series of problems affecting the data transmission of IoT and the efficiency of data processing have been solved. The main problem with IoT data perception is that it is difficult for IoT nodes to continue to operate for a long time, which results in a significant decline in the scalability and usability of their systems. Driven by energy demand, passive sensing technology has developed rapidly and is receiving more and more attention and research.

At present, traditional sensors cannot fully meet the current needs, and some important places and places that are not well maintained require new sensors to be deployed. Therefore, there are some small and ultra-low-power sensors to deal with these environments. These sensors are different from traditional wireless sensor technologies and have different applications. For example, it can be implanted in the animal’s heart to sense biological characteristics [1], and embedded in a wall to monitor structural deformation [2].

This type of sensor node has its advantages over traditional sensing devices. For example, under a very low energy budget. Therefore, ultra-low power microcontrollers are needed. Sleep mode makes power consumption more efficient when the microcontroller is powered by the battery, but the active radio still consumes too much power. Therefore, ultra-low power communication mechanism backscatter communication is widely used.

In real life, in the application scenarios of warehouse management, environmental monitoring, and logistics monitoring, a large number of passive devices will be deployed in a very wide area and transmit a large amount of perceptual data. At this time, the problem of low throughput of passive sensing network has become a main bottleneck for large-scale Internet of Things scenarios. In order to ensure high throughput, research is mainly carried out in three aspects: rate adaptation, parallel transmission, and attention to channel quality.

First, the rate adaptation is carried out in the backscatter communication network, and the transmission bit rate is adaptively selected by the channel quality to improve the system throughput, but the restriction condition is channel quality. Firstly, the backscatter device harvested the Radio Frequency (RF) energy from the reader to transmit the collected data. The low power consumption results in poor signal quality, makeing the communication between the tag and the reader vulnerable to channel quality. Secondly, the environment of the backscatter communication networks is usually complicated, and there are effects such as people walking and other wireless networks. This causes factors such as multipath fading and external effect to affect channel quality. Therefore, in order to ensure high throughput, it may be considered to dynamically adjust the data rate to accommodate channel variations based on channel quality. Starting from this problem, some other issues are raised, such as channel quality estimation, link burst, and rate selection.

Secondly, at present, the scheme for improving throughput is to parallelize multiple backscatter transmissions, concurrent transmission is proposed in the backscatter system, and concurrent transmission can be used for large data transmission, but most concurrent transmission schemes assume that the channel coefficients are unchanged. An important challenge for concurrent transmission is how to decode conflicting signals, and the reader can use its ability to sample at a much higher rate than the tag to separate interleaved signal edges from different tags, depending on time or the stable and distinct characteristics of the signals in the IQ domain. However, the signal characteristics can be highly dynamic and unpredictable. Under various variables, research generally assumes that the channel coefficients are constant. Therefore, channel quality is also a limiting condition for concurrent transmission.

Thirdly, concerning the channel quality, the low power consumption characteristics of the backscattering device result in low signal quality, making the communication between the tag and the reader sensitive to channel quality. The modulation channel has spatial diversity and frequency diversity characteristics in a passive sensing system. Some of the existing methods in backscatter communication networks do not fully consider spatial diversity and frequency diversity. Because the tags are dispersed in different locations, they have different channel quality as a result of many factors, such as channel fading and multipath effects [3]. This phenomenon is expressed as spatial diversity. Further, frequency diversity is the result of the different center frequencies, even for the same node, channel quality is different. This difference is mainly caused by frequency selective fading [3], which means that the frequency response is not monotonic.

Therefore, the research of the three aspects is ultimately the channel quality, and the channel quality can directly affect the research of rate adaptation. The precondition for BLINK [4] is to assume that all nodes experience the same channel quality and simply replace all nodes with the channel state of a single node. Buzz [5] models the network as a single tap channel, ignoring the effects of frequency diversity. In order to improve the above method, CARA [3] proposed a channel-aware rate adaptation method considering spatial diversity and frequency diversity, but it requires a small interval of time to probe channel quality and increases the overhead of channel probing. As the downlink rate also greatly affects the overall throughput, RAB [6] not only considers spatial diversity and frequency diversity, but also selects the overall uplink and downlink rate as the research object, improving the overall throughput. To avoid collisions, it also proposes filter-based probing to efficiently estimate the channel. However, the existing related research has a large estimation cost for the channel, and it is necessary to repeatedly measure the channels at different positions and at different times, and the obtained channel state is not real-time, so the obtained adaptive result cannot achieve the maximum efficiency.

We consider channel diversity and want to reduce the probe overhead. Because the channels are correlated, violent mutations rarely occur, so we use predictive methods to reduce the number of probes. Therefore, this paper proposes a channel prediction method. The overall uplink and downlink rates are used as link indicators while considering spatial diversity and frequency diversity to reduce the probe overhead and achieve real-time prediction. This paper mainly proposes a channel prediction framework based on BP (back propagation) neural network. We set up the monitoring module and prediction modules. The detection module sets the acceleration threshold and link burst threshold, respectively, to monitor whether the channel quality at the current time needs to be re-acquired, and continuous probing is avoided. The prediction module predicts the channel quality of the next moment.

Our contributions are mainly as follows:

1. We propose a prediction framework. The monitoring module is firstly set up to trigger the prediction through the acceleration threshold and the link burst threshold, and then the prediction module uses the neural network algorithm.

2. We use channel prediction to achieve which channel to select and use at the next moment, avoiding continuous detection in passive sensing networks. At the same time, it is possible to predict the quality of all channels in the next stage, which is essentially different from the channel estimation to estimate the quality of the current channel.

3. We confirm the superiority of this prediction mechanism by comparing with previous work. The prediction time can meet the frequency hopping interval, that is, the switching channel time, and the throughput is improved to some extent.

## 2. Overview and Motivation

### 2.1. Physical Layer

In a backscatter communication network, there are many differences between the forward link and the backward link. On the one hand, the encoding mode of the forward link is weak, while the backward link uses a stronger encoding. On the other hand, the forward link receiving antenna is less sensitive and the backward link receiving antenna is highly sensitive. Therefore, owing to the difference in the encoding mode and antenna reception, the forward link path loss is lower and the backward link has higher path loss. The forward link is a reader-to-tag communication link, and a Pulse Interval Encoding (PIE) that is easy to decode is used. However, the backward link uses different encoding modes and baud rates. The encoding methods include the following four types: FM0, Miller2, Miller4, and Miller8, and the baud rate is 32 kbps to 640 kbps. As shown in Table 1, there are six bitrates that are specified by EPC Class-1Gen-2 standard, and the bitrate is produced by different baud ratio and encoding scheme combinations.

### 2.2. Frequency Hopping

A typical Ultra High Frequency (UHF) reader sets up hopping between 50 channels in the 902 MHz~928 MHz ISM band to avoid interference between readers. The frequency of operation of the radio varies from region to region. For example, in China, the frequency is limited to 920 MHz–925 MHz, with 16 channels, 250 kHz per channel. However, the range of operating frequency in North America is from 902 MHz to 928 MHz. To reduce interference in the reader channel, the FCC method specifies the maximum channel duration of 0.4 s in one cycle (10 s). The commercial reader implements a 50 channel sequential hopping frequency with a duration of 0.2 s per channel.

### 2.3. Channel Correlation

As shown in Figure 1, in the EPC protocol, we have 16 channels available. We place tags at the same location in the office environment, use readers to change the channels, and measure the average data read rate in the same time period. There is a correlation between the channels, and adjacent channels usually exhibit similar qualities, as shown in Figure 1, so there is no need to probe all channels. For example, the reading rate of channels 1 to 5 is basically 12 reads/s, and the reading rate of channels 10 to 13 is about 22 reads/s.

The channel quality has little changes with a short time interval, as shown in Figure 2, where channel 13 exhibits excellent stability, which is basically stable at 28 reads/s. But channel 1 and channel 5 undergo significant changes. When channel 1 is used, the node suddenly moves quickly at 14 s, and the reading rate drops to 18 reads/s. Therefore, the channel exhibits good stability in the case of the stable environment, and the channel changes significantly when the environment changes. We combine the correlation between channels and the stability of the channel, which prompted us to consider using predictive methods to select a better channel.

### 2.4. Channel Condition Metrics and Experimental Environment

Generally, the channel quality indicators include detection packets, Signal-noise Ratio (SNR), Channel State Information (CSI), long-term statistical rate, and so on, but for passive sensing systems, the channel detection indicators are limited, and owing to the changing characteristics of wireless channels and spatial diversity, the channel state changes in real time.

Commercial Radio Frequency Identification (RFID) readers expose two link metrics: (a) the Received Signal Strength Indicator (RSSI)value for each query response from a sensor tag, and (b) the aggregate per-channel loss rate for each dwell time interval. Meanwhile, a unique feature of backscatter communication is that packet loss and RSSI provide complementary information about path-loss and self-interference, and thus need to be used in conjunction. They also affect the reading rate.

Among them, RSSI and reading rate can be directly obtained through commercial RFID readers, and packet loss rate can also be obtained through simple calculation. The Bit Error Ratio (BER) needs to be obtained through calculation. In our research, we only consider whether the data packet is received or not, which is measured by the packet loss rate, so we do not use the BER metric. SNR cannot be obtained directly in the backscatter network, it needs to be obtained through professional equipment, so we do not use the SNR metric.

Our stable and unstable environment is based on gesture recognition [7] and tracking [8] mentioned that human activity will affect the phase and RSSI, and RSSI is the strength of the reception, which characterizes the channel state information. Therefore, we carried out previous work research. We placed a reader and a tag with a distance of 1.5 m. We chose fixed channel 5 for communication without interference from other signals of the same frequency. In the office environment, the phase and RSSI values were measured without people walking. As shown in Figure 3a, the RSSI is almost stable at −21 dBm, so we define it as a stable environment. Under the same conditions, two people are walking within the communication range of the reader and the tag at the same time. The experimental results are shown in Figure 3b. When the two people are between the reader and the tag at the same time, the RSSI value is different by 12 dBm. Therefore, we define an environment in which at least two people move around as an unstable environment.

## 3. System Overview

Figure 4 shows the framework of channel prediction. As Computational Radio Frequency Identification (CRFID) is usually embedded in or attached to an object, the movement of the object itself, and the impact of people walking around, these two states will affect the quality of the channel. So, when we set up the monitoring module, we use the acceleration sensor to monitor the motion of the object itself, and use the link burst metric to determine the changes in the surrounding environment. We can use two monitoring indicators to determine whether the channel quality has changed.

When the monitoring module finds that the channel quality changes, it needs to re-probe the channel quality. The prediction module re-predicts the channel quality using the existing channel quality indicator according to the channel correlation. The primary responsibility of this module is to predict and select channels to maximize throughput. We determine the quality of the channel by the characteristics of the channel, such as RSSI, packet loss rate, throughput, and so on. The algorithm of the neural network is mainly used to predict the list of preferred channels using the known data and the correlation between the data. The prediction algorithm predicts the RSSI, packet loss rate, and reading rate of all channels at the next moment. We mark the three channels with the best RSSI, the best packet loss rate, and the best reading rate, and output them as the best channel, that is, the first three channels in the preferred channel list.

## 4. Channel Prediction Design

### 4.1. Link Burstiness

The inherent dynamic variability and unpredictability of wireless links mean that real-time and accurate link quality estimation still face huge challenges, especially in complex network environments, such as moving people around, various static or mobile obstacle, other wireless networks, electronic devices, and so on, and the fact that the state of the wireless link changes rapidly in a short time, exhibiting a high degree of burstiness.

Conditional packet de-livery functions (CPDFs) improves the success and failure of packet length vector transmission to briefly represent burstiness. The Kantorovich–Wasserstein (KW) distance [9] and the conditional packet transfer function are combined to express the burstiness as a single scalar. Therefore, this method is used to measure the proximity of CPDF to an ideal burst link. In other words, the KW distance between two vectors is the average of the absolute differences of the corresponding elements of the two vectors. The burstiness metric β [10] is defined as follows:(1)$$\beta =\frac{KW\left(I\right)-KW\left(E\right)}{KW\left(I\right)},$$
where E is the CPDF of the empirical link in question, and I is the CPDF of the independent link with the same packet reception ratio.

Figure 5 plots the link burstiness of the channels we observe at different locations, and the loss rate of these channels is neither 100% nor 0%. β = 1 indicates that the link is bursty, and β = 0 indicates that there occurs an independent packet loss. We can observe that these channels exhibit high burstiness when β > 0.75. Therefore, we set up a threshold for the link burst metric as 0.75. When β > 0.75, it indicates that the link is bursty and the surrounding environment changes. We need to measure the link indicator at the moment to retrain and learn. Thus, we can obtain the predicted channel under environmental changes.

### 4.2. Mobility Detection

When the sensor moves, the channel characteristics change. In order to maximize throughput, the reader needs to change the encoding, baud rate, or channel [11]. In this section, we present a method to determine the sensor tags’ mobility.

BLINK and CARA have the same module. BLINK monitors some mobile indicators, such as the RSSI vector distance, while CARA mainly targets position movement. The mobility of the sensor cannot be well described by simply using the phase or some RSSI vector distance. Therefore, we use acceleration sensor data as an indicator of mobility detection.

We define the type ID in order to classify the different values of the sensor data. The type of sensor acceleration (standard) is 0 × 0 D. The type of sensor acceleration (quick) is 0 × 0 B. The reader collects accelerometer values in a 12-byte EPC format when moving before the antenna of the RFID reader. This EPC format includes 1-byte tag type, 8-byte data, 1-byte WISP (wireless identification and sensing platform) hardware version, and 2-byte hardware serial [12]. By default, the WISP returns a sensor type ID, sensor data (optional), and a tag ID that encodes the firmware version. If sensor data are included, 7 bytes will be set aside for the sensor payload, followed by tag ID. The data will start in the PC field (two bytes in the ACK response), and spread into the EPC field (8 bytes in the ACK response).

We can get the acceleration value by the following formulas:(2)$$Accel_{x},Accel_{y}=\left(100-100\times 1.16\times value\_returned\_by\_wisp\right)/1024,$$
(3)$$Accel_{z}=\left(100\times 1.16\times value\_returned\_by\_wisp\right)/1024,$$
where $Accel_{x,y,z}$ is the value of triaxial accelerometer.

When the sensor moves, the value of the triaxial accelerometer will definitely change, so we can judge whether it is static or mobile as follows:(4)$$B_{t}=\left|Accel_{x}+Accel_{y}+Accel_{z}\right|,$$
where $B_{t}$ is the modulus of the vector formed by the triaxial accelerations.

Figure 6 plots the relationship between acceleration data and channel read rate. According to previous experimental experience, the reading rate of the sensor node below 16 reads/s is a poor reading rate. As shown in Figure 5, when the reading rate is 16, the modulus of the triaxial accelerations is 5.1, so we set the threshold value to 5.1 based on experimental measurements and experience. We found that, when the modulus of the triaxial accelerations is bigger than 5.1, the channel quality is bad for read data, so we set up a threshold for $B_{t}$ as 5.1. When $B_{t}$ > 5.1, this means the sensor node moves, and the channel quality is greatly affected, so we re-measure the channel quality and perform new channel prediction.

### 4.3. Channel Quality Prediction

In wireless communication, in order to effectively utilize the limited wireless spectrum, how to improve spectrum utilization is currently the main problem. As the mobile channel is time-varying, the dynamic range of its channel variation is very large. The traditional fixed-mode modulation scheme is designed in the channel condition with the worst channel condition or average channel state, thus losing the system efficiency. The adaptive modulation that changes the modulation mode according to the channel state has attracted great attention. For backscatter communication in passive sensing systems, the necessity of channel prediction is as follows. First, accurate real-time channel prediction can be better adaptively modulated, thereby improving spectrum utilization. Second, after the channel prediction, direct selection of better channel transmission information makes better use of the spectrum than changing the modulation scheme.

Channel estimation all uses known sequences, traditional channel estimation algorithms include common RLS, LMS, MMSE, and compressed sensing channel estimation, including MP and OMP in the greedy algorithm, and LS0, LS0-BFGS, and LS0-FR in the convex optimization algorithm. Existing methods for backscattering systems usually assume that the channel state information is well known at the transmitter and the channel estimation is faultless at the receiver. In practical systems, these assumptions are not available because, during the time delay of estimation and data transmission, the time-varying characteristics of the wireless channel cause the state of the channel to change. Therefore, selecting the appropriate modulation mode according to the channel state will lead to the degradation of system performance. Because the existing methods have the drawback of time delay, we propose a real-time prediction method to minimize the delay to improve the system performance and reduce the probe overhead.

Compared with the traditional channel estimation algorithm, the BP neural network takes a large number of known channels as input, and can learn and adapt to the dynamic characteristics of the channel with faster change and correlation, with higher prediction accuracy. At the same time, the traditional algorithm cannot realize real-time prediction as a method of channel estimation. BP neural network can set the input layer, hidden layer, and output layer for short-term real-time prediction. The various neurons of the network can store all quantitative or qualitative information, and the network is highly robust and fault tolerant. The time complexity of the BP neural network algorithm can be adjusted by adjusting the number of input neurons and the number of hidden layers. At the same time, traditional channel estimation can only estimate the state of the current channel at the next moment. The use of neural network prediction can predict the state of all channels at the next moment, which is conducive to our choice of better channels for transmission.

In order to obtain accurate channel quality metrics, it is necessary to probe the channel. Because there is spatial and frequency diversity, we want to obtain metrics for each channel and each node. However, as the number of channels increases and the accuracy requirements increase, the probing overhead will become higher. BLINK does not consider spatial and frequency diversity. Only one probe is used. First, it cannot reflect the channel quality in fine granularity. Second, as the number of nodes increases, the problem of collision cannot be solved. CARA uses selective probing, using enough probes to make full use of channel correlation to reduce probing overhead. Using channel correlation, we propose a prediction mechanism based on neural networks. The channel quality will change strongly when the environment changes, so we use the prediction mechanism to replace the method of probing at intervals.

We use the prediction algorithm based on BP neural network to predict the next moment channel quality. This design uses a BP neural network with a hidden layer, that is, the number of neural network layers is 11. Two models were established for two different environmental predictions.

For the first input of the input layer, we construct an m × n matrix C, m and n are the available channels and the number of nodes, respectively. In C, $C_{\mathrm{ij}}$ represents the reading rate of the node j at channel i; the second input is the m × 1 matrix A, indicating the RSSI of each channel; the third input is the m × 1 matrix B, indicating packet loss rate per channel. The output data are a predicted matrix containing the read rate, RSSI, packet loss rate, and a list of preferred channels. The above input does not represent the input format, because the input of BP neural network can only be a scalar. We input according to the reading rate, RSSI, and packet loss rate of each channel. Therefore, the number of input layer and output layer nodes is 16 and 4, respectively. The number of hidden layer nodes is determined according to the empirical Formula (5).
(5)$$y=\sqrt{l+q}+a,$$
where l is the number of input layer nodes, q is the number of output layer nodes, a is a constant between 1 and 10, y is the number of hidden layer nodes, and thus the value range of y is 6 to 15. The number of hidden layer nodes obtained by experiments is nine.

The transfer function usually uses the Sigmoid function. After experimentation, the transfer function of the hidden layer is determined as follows:(6)$$y=\frac{2}{1+\exp(-2n)}-1.$$
The weight learning method is the gradient descent method with additional momentum terms.

The predicted data reading rate, RSSI, and packet loss rate were taken as the latest measured data and supplemented as training data. The weights were adjusted for each prediction to achieve real-time dynamic prediction.

## 5. Implementation

In this part, we put forward implementation issues. The overall hardware experimental platform is shown in Figure 7. Universal Software Radio Peripheral (USRP) N210 equipped with SBX-40 daughter board can be used as a detector and reader. This article uses the open source written by Nikos on GitHub to use USRP as a reader, and on this basis, make appropriate modifications based on experimental conditions and design. The SBX-40 daughter board provides MIMO functionality and provides 40 MHz bandwidth. The operating frequency is 400 MHz Up to 4400 MHz.

The platform used in the experiment is 64-bit Ubuntu 14.04 and GNU Radio 3.7.4. The selected CRFID tag is the WISP4.1 version with a dipole antenna, and its microcontroller is MSP430F2132. The commercial reader model used is ImpinJ Speedway R420, which is connected to the Laird circularly polarized directional antenna S9028PCL. It can connect up to four antennas at the same time, and the antenna gain is 9 dBi.

We use Impinj Speedway reader and the WISP as backscatter nodes in the network. This paper sets up experimental points in the office, placing sensor nodes in different locations. A stable environment is an office with no people moving around, and an unstable environment is an office with two people moving around. It is worth noting that the positions and number of sensor nodes are the same in stable and unstable environments. Further, we used USRP software radio reader to implement our framework.

We use multiple tags to measure the RSSI, packet loss rate, and reading rate of a channel in the current environment at the same time. Then, we switch channels to obtain the same channel index. Then we store these data for training and learning. Our experimental test scenarios are shown in Figure 8. We placed a reader and tags with a distance of 1.5 m without interference from other signals of the same frequency. Figure 8a shows that multiple tags are placed in a stable environment, and Figure 8b shows tags placed at the same location in an unstable environment when two people walk around.

**Training data:** First of all, our training data are obtained in a stable environment and unstable environment. A stable environment refers to a situation where the position of the node is unchanged and there is no interference in the surrounding environment, and the unstable environment refers to a situation where nodes move or there are multiple people walking around. The data set we need to obtain includes the RSSI of the channel, the packet loss rate, and the data read rate. Data were collected for 20 min in a stable environment and 30 min in an unstable situation. Because of the large difference in data in an unstable environment, we obtain more data in an unstable environment than in a stable environment.

**Predicted and monitoring time:** It is worth noting that the maximum dwell time for a single channel varies by region. For example, in North America, the FCC allows a single channel to be up to 0.4 s in 10 s. In Europe, if the tag is being read, only 4 s are allowed to be transmitted on a single channel, and if no tag is read, only 1 s is allowed. In China, the maximum dwell time is 2 s. Therefore, we set our prediction time to 2 s, which means that re-prediction occurs every 2 s. Because the channel quality is relatively stable and there is no other impact in a stable environment, in the unstable environment, more training data are needed, so the channel quality monitoring time is set to 1 min.

## 6. Evaluation

In this section, we evaluated our framework through experimentation and simulation. In the experiment, we used Impinj Speedway reader and passive CRFID as nodes. The evaluation consists of three parts: (1) measuring the predictive effect of our algorithm; (2) evaluating the real-time performance of the prediction algorithm; and (3) evaluating and comparing the overall framework with the throughput of BLINK and CARA in different environments.

### 6.1. Predictive Effect

We evaluated the effects of our prediction algorithm, using simulation experiments and five tags. In order to obtain experimental data, we sequentially read the data of each channel with a reader, and obtained the RSSI and packet loss rate of each channel. We compared our prediction algorithm with the channel quality under real conditions. The real value is selected by random hopping, that is, the channel is randomly selected every 2 s. We compared them in a stable environment and an unstable environment. The comparison results are shown in Figure 9.

Although random hopping can avoid poor channels, it is unstable because of randomness. By default, the commercial reader performs frequency hopping every 2 s. The purpose of frequency hopping is to prevent a situation where the channel throughput is poor, resulting in a decrease in system throughput or even a communication failure.

In the case of stability, the traditional commercial reader frequency hopping communication is compared with the channel prediction algorithm. Traditional frequency hopping communications also maintain a good read rate owing to comparisons in a stable environment. As shown in Figure 9a, our algorithm is similar to the actual effect. The actual reading rate we measured is between 18 and 28 reads/s, and the predicted reading rate is stable between 25 and 28 reads/s. Because the channel quality is stable in stable conditions, the prediction algorithm does not show superiority.

In an unstable environment, this paper sets up the experimental site in the office and placed five tags in five different locations. In the office environment, there are more complex multipath effects and irregular movement of people, which has a huge impact on the communication stability of passive sensing systems. In the reading range of the reader, five different positions of the tags are arranged, and the reader collects sensing data. In this paper, the reader collects data for 31 min, uses the channel prediction algorithm to perform 30-min actual scene training, compares the remaining 1 min actual data with the predicted 1 min data, and judges its prediction effect.

The result is shown in Figure 9b; the actual reading rate we measured is between 10 and 23 reads/s, and the predicted reading rate is stable between 22 and 27 reads/s. It can be seen from the figure that, when the environment around the tags changes, for example, if there is a person moving in the office, the communication process will have a greater impact, and the reading rate will also have large fluctuations. Such communication is poor in stability. Unstable throughput cannot meet current communication requirements.

In contrast, a better data transmission rate can be achieved by using the channel prediction method. When the surrounding environment changes, the current channel state also changes. As a result, the original communication channel may be unable to adapt to the current environment, resulting in a change in the data reading rate. If the communication channel is not changed in time, there will be a case where the data read rate suddenly drops to 10 reads/s and the throughput is affected in Figure 9b. Channel prediction can change the communication channel in time to adapt to changes in the current environment. Channel prediction can effectively maintain the stability of the passive sensing system.

We set up a tag and a reader in a stable and unstable environment, respectively, and select a fixed channel 13. Because the prediction algorithm does not consider channel selection, frequency hopping is not required. We select the RSSI prediction results within 60 s, as shown in Figure 10. Because the RSSI value can be obtained in real time, in the verification of the prediction effect, the actual RSSI value is the average value every 2 s. The prediction result is −21 dBm in a stable environment because the actual RSSI changes are small. The prediction results in the unstable environment are roughly the same as the actual results, which reflects the accuracy of the prediction.

We also performed a prediction of the packet loss rate at the same time. The prediction result is shown in Figure 11. In a stable environment, the prediction result of the packet loss rate is similar to the actual result, and both are stable between 0% and 20%. In an unstable environment, as people move around, the packet loss rate changes significantly. When the impact is severe, the actual packet loss rate will increase to 99%. Our prediction algorithm can also predict certain results.

### 6.2. Real Time

We evaluated the real-time nature of the prediction algorithm; because channel changes are real-time, real-time performance is important. However, the maximum dwell on a single channel is 2 s, so our prediction real-time is less than 2 s to make an effective channel selection. The experimental results are shown in Figure 12, and real-time performance was tested in a stable and unstable environment, respectively. The measurement time is the first training time, and the measurement accuracy can only be accurate to the millisecond.

As shown in Figure 12a, in a relatively stable environment, the prediction is made every 2 s. Because there is a well-trained model before, in a stable environment, the prediction time is extremely short, and it can be effectively predicted. Further, as the number of tags increases, the training time stays within 0.41 s.

The unstable environment is shown in Figure 12b. Every 2 s is predicted, and it needs to be re-learned, so it takes more time. However, the channel dwell time is 2 s, so our real-time performance is reliable. The increase in the number of tags will increase the time for re-learning, so the change will be more obvious. When the number of tags reaches 20, the prediction time reaches 1.6 s, which is less than 2 s and meets the frequency hopping interval.

### 6.3. Overall Performance

We evaluated and compared the previous work BLINK and CARA with our overall framework. As the Impinj Speedway reader cannot fully implement our framework, we evaluate with a simulation. We used 5 tags, 10 tags, 15 tags, and 20 tags for comparison, and we place tags in two different environments, a stable and an unstable environment.

Figure 13 shows a comparison of the three ways in a stable environment. When the number of placed tags is five, our overall throughput reaches 85 reads/s, which achieves 1.18× throughput gains over BLINK and CARA. When the number of placed tags is small, the difference of throughput is not very large, because channel diversity is not fully reflected when the tags’ number is small. We use the rate matching method in the CARA framework, so it does not show a large increase. When the number of tags reaches 20, the overall throughput reaches 43 reads/s, while the throughput of both CARA and BLINK is 35 reads/s, which is a 1.23× gain. When the number of placed tags is large, spatial diversity and frequency diversity are reflected, and channel prediction also shows an advantage. Although the method of rate selection in CARA is used, the prediction algorithm has a certain improvement in throughput compared with CARA.

Figure 14 shows that we placed a different number of tags in an unstable environment. When there are five tags, the throughput of our method is 0.5× and 1.4× better than BLINK and CARA, respectively. The same trend can be observed across different tag numbers. The throughput of the three methods is significantly reduced, but CARA is significantly better than BLINK, and our prediction method is better than CARA. All three methods have mobility monitoring modules, but we have increased the monitoring of link burstiness, so the prediction method shows its superiority in an unstable environment.

## 7. Related Work

There has been a lot of prior work on wireless channel characteristics and link layers. Much of the work on backscatter communication is to optimize EPC Gen 2 tag communication, for example, better tag collision avoidance [13], better security [14], collect IDs of an unknown set [15], efficient packet protocol [16], and reduced inventory time [17]. Some work analyzes the main propagation characteristics of the backscatter system by using four link parameters [18]. Other work focuses on energy harvesting issues, such as the collaborative approach to WISP device charging in wide-area sensor networks [19], and collecting energy from the environment to charge nodes [20]. None of these solve the link layer adaptation mechanism to improve throughput.

Previous work optimized for backscatter communication links is not compatible with C1G2. BUZZ [5] uses the collision information as a ratless code to dynamically adjust the bitrate for lossless transmission. Flit [21] improves the MAC layer of the protocol, enabling fast transmission of bulk data and reducing transmission time. Laissez-Faire [22] proposed decoding parallel transmission by analyzing signals in the time domain and IQ domain, but it is susceptible to channel quality. BiGroup [23] also performs parallel decoding on the IQ domain, but it combines the transition probability with the IQ domain to create a physical model that greatly improves throughput. Although these C1G2 incompatible optimizations have a significant performance improvement, they are still not suitable for large-scale readers and nodes.

Later, some improvements compatible with the C1G2 protocol were proposed. The precondition for Blink [4] is to assume that all nodes experience the same channel quality, using specific calculations to monitor mobility and rate selection. CARA [3] observed an opportunity to increase throughput through channel-aware rate selection. Because the downlink rate can also greatly affect the overall throughput, RAB [6] focuses more than on the choice of uplink rate selection, which is for overall throughput. In order to avoid collision, it also proposes filter-based probing to effectively estimate the channel.

## 8. Conclusions

This paper proposes a framework about channel prediction that considers both spatial diversity and frequency diversity. The threshold of the data of the acceleration sensor and the link burst metric are set, thereby obtaining the retraining moment. Through a certain amount of experiments, we find that the prediction framework has a certain improvement in throughput compared with the previous work.

## Figures and Tables

**Figure 1 sensors-20-00300-f001:**
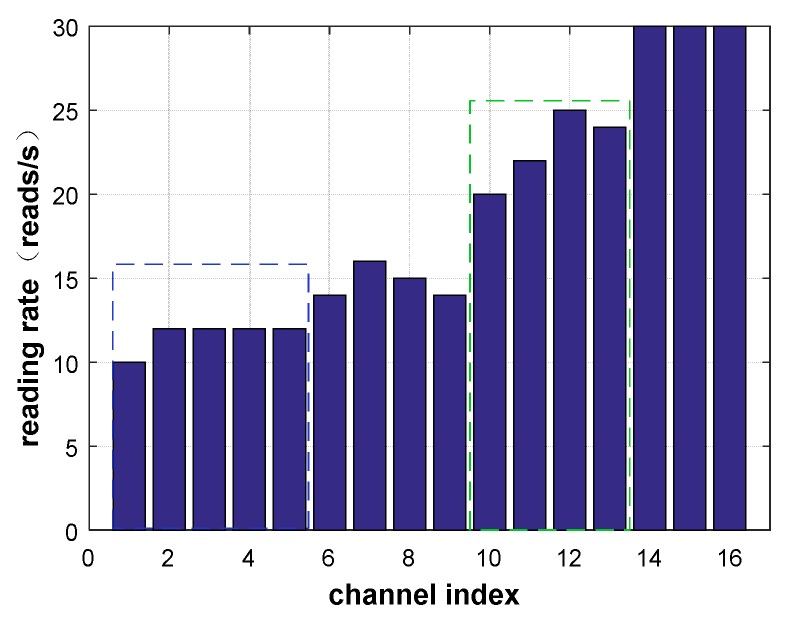
A typical example of channel correlation.

**Figure 2 sensors-20-00300-f002:**
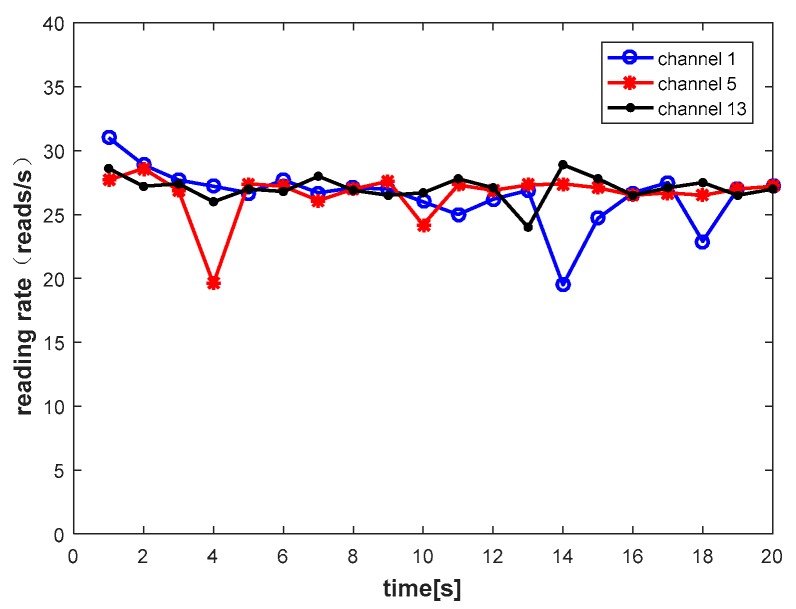
A typical example of channel stability.

**Figure 3 sensors-20-00300-f003:**
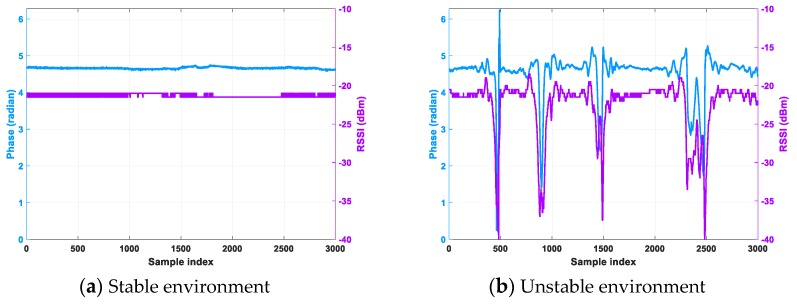
Phase and RSSI in stable and unstable environments.

**Figure 4 sensors-20-00300-f004:**
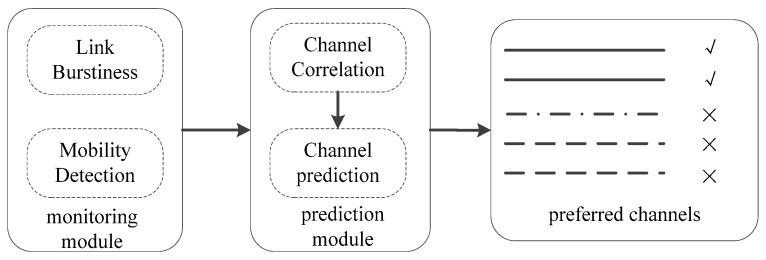
Framework overview.

**Figure 5 sensors-20-00300-f005:**
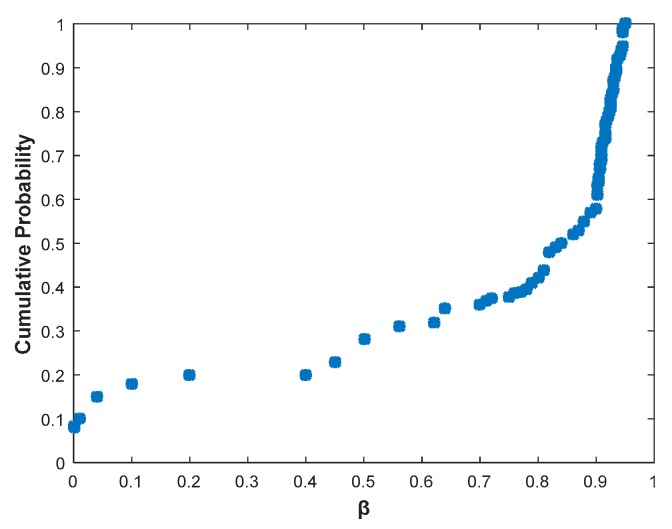
The β value for the links.

**Figure 6 sensors-20-00300-f006:**
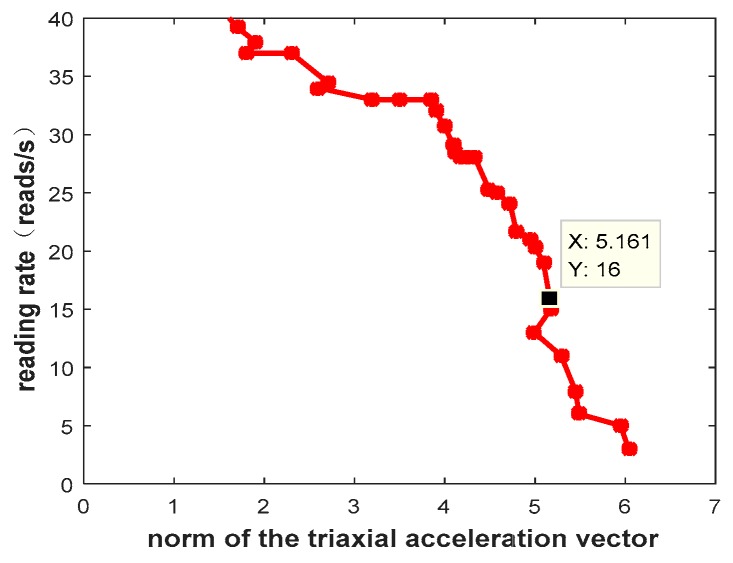
The relationship between norm of the acceleration vector and reading rate.

**Figure 7 sensors-20-00300-f007:**
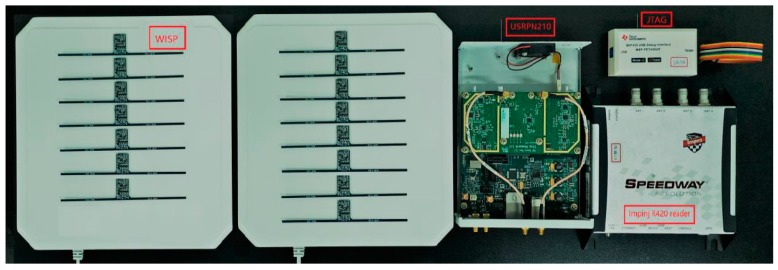
Laboratory equipment.

**Figure 8 sensors-20-00300-f008:**
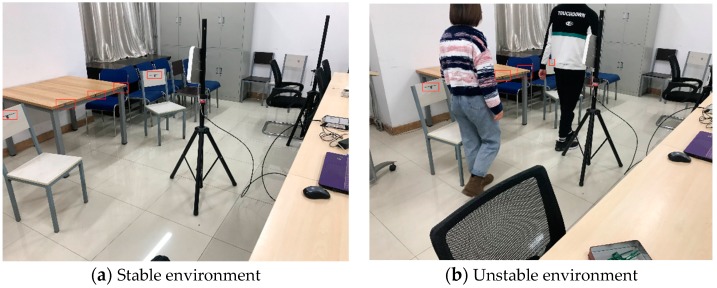
Experimental test scenarios.

**Figure 9 sensors-20-00300-f009:**
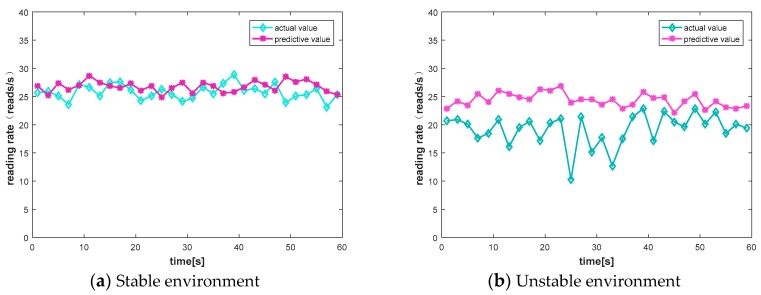
Predictive effect of reading rate.

**Figure 10 sensors-20-00300-f010:**
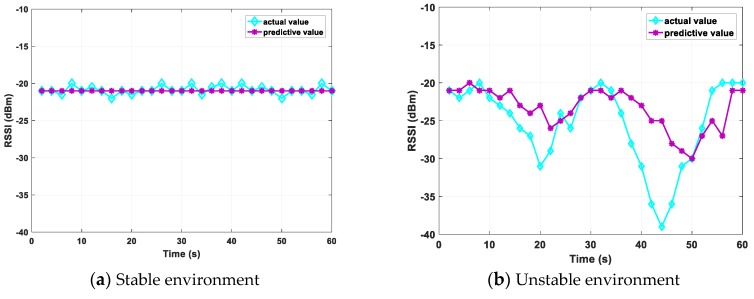
Predictive effect of RSSI.

**Figure 11 sensors-20-00300-f011:**
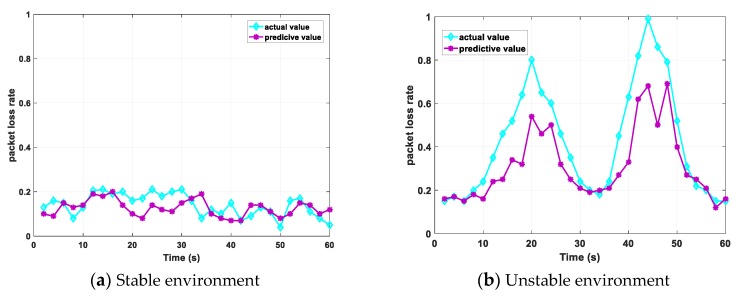
Predictive effect of packet loss rate.

**Figure 12 sensors-20-00300-f012:**
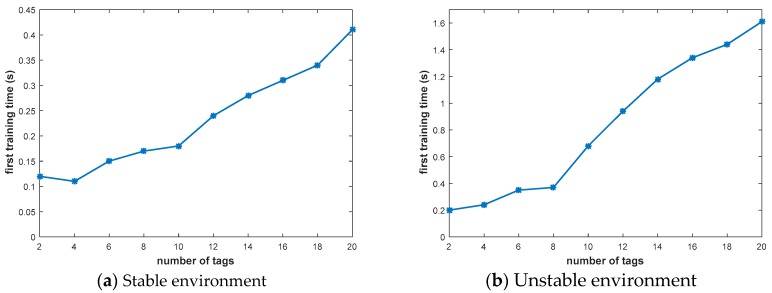
Real time.

**Figure 13 sensors-20-00300-f013:**
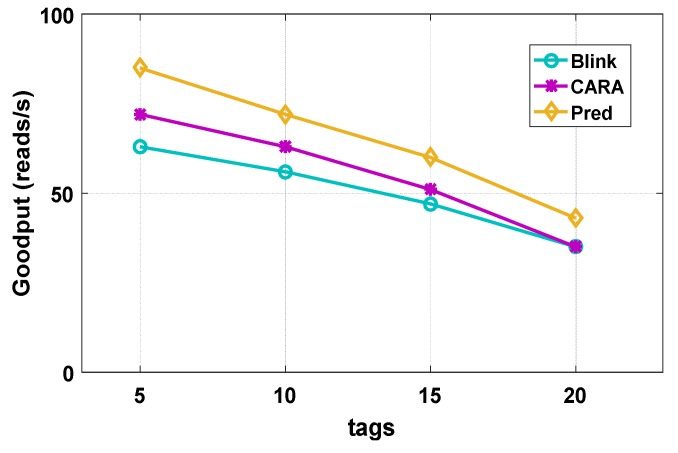
Stable environment.

**Figure 14 sensors-20-00300-f014:**
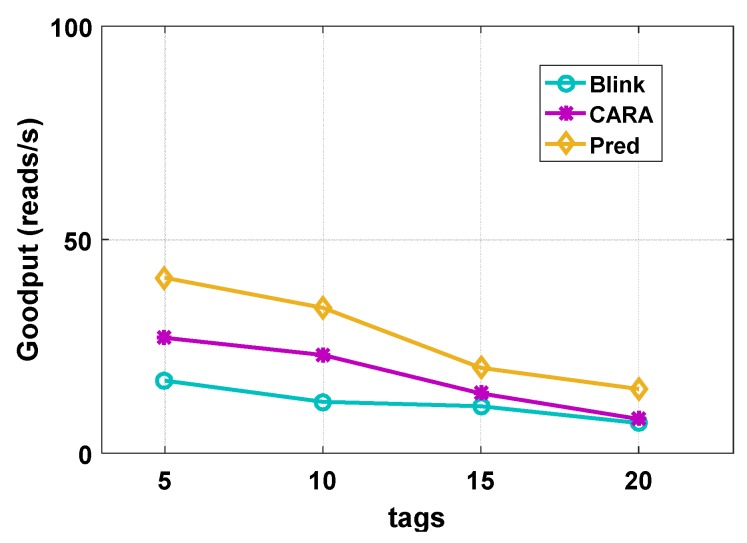
Unstable environment.

**Table 1 sensors-20-00300-t001:** Backward link six bitrate.

Bitrate (Symbols/s)
FM0/64
FM0/160
FM0/40
Miller4/640
Miller4/256
Miller8/256

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
