# Peer review of "Channel Prediction Based on BP Neural Network for Backscatter Communication Networks"

_sensors, 2020, doi:10.3390/s20010300_

Round 1

Reviewer 1 Report

Aiming at the problem that parallel transmission and rate adaptation are limited by channel quality, this paper proposes a channel prediction scheme for backscattering networks by considering spatial diversity and frequency diversity. It is a novel idea to use the acceleration sensor and link burst indicators to trigger channel prediction. The scheme not only reduces the overhead of continuously detecting channels, but also has higher accuracy of channel prediction and improves network throughput.

This paper proposes a channel prediction framework, which is more advanced. Compared with the research in the traditional passive sensing network, it not only solves the channel limitation problem, but also improves the throughput.

Some suggestions for better presentation:

The references in this paper refer to the relevant literature before 2018, and it is recommended to join the discussion of related articles in 2019. Pay attention to formatting issues, such as formulas should keep the font consistent. Section 2.4 should avoid detailed acceleration value acquisition and can be removed from the text. 

Author Response

Point 1: The references in this paper refer to the relevant literature before 2018, and it is recommended to join the discussion of related articles in 2019.

Response 1: Thanks very much for your valuable comments. Here are our updated references:

[10] Ping W , Zhiping Z , Jing L I . Improved Server-less RFID Security Authentication Protocol[J]. Journal of Frontiers of Computer Science & Technology, 2018.

[11] Hou Y , Zheng Y . PHY-Tree: Physical Layer Tree-Based RFID Identification[J]. IEEE/ACM Transactions on Networking, 2018, PP(99):1-13.

[13] Zhao J, Wu H, Li D, et al. LILAC: computable capabilities based high performance protocol for CRFID[J]. IET Communications, 2019, 13(10): 1348-1355.

[16] Alsharif M H, Kim S, Kuruoğlu N. Energy Harvesting Techniques for Wireless Sensor Networks/Radio-Frequency Identification: A Review[J]. Symmetry, 2019, 11(7): 865.

Point 2: Pay attention to formatting issues, such as formulas should keep the font consistent.

Response 2: Thanks for your opinion very much! We modified the paper according to your opinions. Please check it!

Point 3: Section 2.4 should avoid detailed acceleration value acquisition and can be removed from the text.

Response 3: Thanks very much for your valuable comments. We carefully proofread the paper and deleted section 2.4. I hope the current version would meet your requirements.

Reviewer 2 Report

This paper is in general well written and well organized. I only have minor comments as follows: 

A. There are quite a few editing and/or grammatical errors that have to be fixed before I can recommend it for publication. For example,
1) Line 32, duplicate wording in "places and places"
2) Line 35, "traditionally"
3) Line 55, "there are some people walking interference"
4) Line 67,68, the sentence "While the signal..." is incomplete.

B. Please try to be specific in describing the contributions of the paper in Section 1. The authors simply describe what they have done, which is generally insufficient. Contributions should be related to the impacts of the proposed approach on advancing researchers' understanding in the considered problem.

C. Some of the math symbols have strange appearance, e.g. \beta before and after equation (1). Spell out BP in full on its first appearance in the article. What is the rationale behind using NN? Please provide comparisons, such as advantages/drawbacks, between the conventional method and the proposed approach.

Author Response

Point 1: There are quite a few editing and/or grammatical errors that have to be fixed before I can recommend it for publication. For example,

1) Line 32, duplicate wording in "places and places"

2) Line 35, "traditionally"

3) Line 55, "there are some people walking interference"

4) Line 67,68, the sentence "While the signal..." is incomplete.

Response 1: Thanks very much for your valuable comments. We proofread the paper carefully. I hope the current version would meet your requirements.

Point 2: Please try to be specific in describing the contributions of the paper in Section 1. The authors simply describe what they have done, which is generally insufficient. Contributions should be related to the impacts of the proposed approach on advancing researchers' understanding in the considered problem.

Response 2: Thanks for your opinion very much! We modified the paper according to your opinions. The modified content is as follows:

Our contributions are mainly as follows:

We propose a prediction framework. The monitoring module is firstly set up to trigger the prediction through the acceleration threshold and the link burst threshold, and then the prediction module uses the neural network algorithm. We use channel prediction to achieve which channel to select and use at the next moment, avoiding continuous detection in passive sensing networks. At the same time, it is possible to predict the quality of all channels in the next stage, which is essentially different from the channel estimation to estimate the quality of current channel. We confirms the superiority of this prediction mechanism by comparing with previous work. The prediction time can meet the frequency hopping interval, that is, the switching channel time, and the throughput is improved to some extent.

Point 3: Some of the math symbols have strange appearance, e.g. \beta before and after equation (1). Spell out BP in full on its first appearance in the article. What is the rationale behind using NN? Please provide comparisons, such as advantages/drawbacks, between the conventional method and the proposed approach.

Response 3: Thanks very much for your valuable comments.

1) The math symbol \beta  is an index that defines link burstiness, and his expression refers to the link burstiness first proposed by the article [21].

[21] Srinivasan K , Kazandjieva M A , Agarwal S , et al. The β-factor: Measuring wireless link burstiness[C]// Proceedings of the 6th International Conference on Embedded Networked Sensor Systems, SenSys 2008, Raleigh, NC, USA, November 5-7, 2008. DBLP, 2008.

2) When BP first appeared in the article, we had spelled it out completely. Please check it!

3) As we wrote in 4.3, compared with the traditional channel estimation algorithm, the BP neural network takes a large number of known channels as input, and can learn and adapt to the dynamic characteristics of the channel with faster change and correlation, with higher prediction accuracy. At the same time, the traditional algorithm can’t realize real-time prediction as a method of channel estimation. BP neural network can set the input layer, hidden layer and output layer for short-term real-time prediction. The various neurons of the network can store all quantitative or qualitative information, and the network is highly robust and fault tolerant. And the time complexity of the BP neural network algorithm can be adjusted by adjusting the number of input neurons and the number of hidden layers. At the same time, traditional channel estimation can only estimate the state of the current channel at the next moment. The use of neural network prediction can predict the state of all channels at the next moment, which is conducive to our choice of better channels for transmission.

Reviewer 3 Report

I don't see correlations of the reading rate with RSSI, BER, SNR (distance), packet loss, and burstiness metric. You should explain why these metrics are absent from your measurements.

There is no information regarding the spectrum: how it was measured and how do you select the channels based on your methodology (See Figure 3). There is no explanation correlated with measurements.

Regarding acceleration, you have presented the sensor's output dependence. If the modulus of acceleration is your chosen norm, then it is not relevant for measuring the distance (Figure 5). You have established a threshold of 5.1, but there is no explanation.

Regarding the unstable and stable environments, there is no information (raw data) about their characteristics. 

The experimental settlement is not detailed accordingly. The measurements should be clearly presented in order to highlight the selected methodology and the obtained results.

Author Response                                                                                             

Point 1: I don't see correlations of the reading rate with RSSI, BER, SNR (distance), packet loss, and burstiness metric. You should explain why these metrics are absent from your measurements.

Response 1: Thanks very much for your valuable comments.

        Generally, the channel quality indicators include detection packets, SNR, CSI, long-term statistical rate, etc., but for passive sensing systems, the channel detection indicators are limited, and due to the changing characteristics of wireless channels and spatial diversity, the channel state changes in real time.

        Commercial RFID readers expose two link metrics: a) the RSSI value for each query response from a sensor tag, and b) the aggregate per-channel loss rate for each dwell time interval. Meanwhile, a unique feature of backscatter communication is that packet loss and RSSI provide complementary information about path-loss and self-interference, and therefore need to be used in conjunction. They also affect the reading rate.

        Among them, RSSI and reading rate can be directly obtained through commercial RFID readers, and packet loss rate can also be obtained through simple calculation. The BER needs to be obtained through calculation. In our research, we only consider whether the data packet is received or not, which is measured by the packet loss rate, so we do not use the BER metric. SNR cannot be obtained directly in the backscatter network, it needs to be obtained through professional equipment, so we do not use SNR metric.

Point 2: There is no information regarding the spectrum: how it was measured and how do you select the channels based on your methodology (See Figure 3). There is no explanation correlated with measurements.

Response 2: Thanks for your opinion very much!

1)We use multiple tags to measure the RSSI, packet loss rate, and reading rate of a channel in the current environment at the same time. Then we switch channels to obtain the same channel index. Store this data for training and learning.

2) Our prediction algorithm predicts the RSSI, packet loss rate, and reading rate of all channels at the next moment. By marking the best RSSI, the best packet loss rate, and the best reading rate, the three best channels are obtained. Subsequent channel selection is selected among these three channels. If the condition is still not met, then try to find other channels. Our follow-up research is studying better channel selection methods. If the predicted channel cannot reach the best, we will change how to deal with it. This is our future work.

Point 3: Regarding acceleration, you have presented the sensor's output dependence. If the modulus of acceleration is your chosen norm, then it is not relevant for measuring the distance (Figure 5). You have established a threshold of 5.1, but there is no explanation.

Response 3: Thanks very much for your valuable comments.

1) Because we use WISP (Wireless Identification and Sensing Platform) as a sensor node, and WISP has a limited communication distance compared to ordinary tags without computable functions, we did not take the distance into consideration when setting the threshold. The threshold we set is not related to the distance. Figure 5 only shows the relationship between the modulus of the triaxial accelerations and the reading rate.

2) According to previous experimental experience, the reading rate of the sensor node below 16 reads/s is a poor reading rate. As shown in Figure 5, when the reading rate is 16, the modulus of the triaxial accelerations is 5.1, so we set the threshold value to 5.1 based on experimental measurements and experience.

Point 4: Regarding the unstable and stable environments, there is no information (raw data) about their characteristics.  

Response 4: Thanks for your opinion very much! We set up experimental points in the office, placing sensor nodes in different locations. A stable environment is an office with no people moving around, and an unstable environment is an office with people moving around. It is worth noting that the positions and number of sensor nodes are the same in stable and unstable environments. At the same time, Figure 7 and Figure 8 show the raw data of the reading rate in the stable environment and the unstable environment, where the reading rate in the unstable environment varies greatly.

Point 5: The experimental settlement is not detailed accordingly. The measurements should be clearly presented in order to highlight the selected methodology and the obtained results.

Response 5: Thanks very much for your valuable comments.

        The overall hardware experimental platform is shown in Figure 6. USRP N210 equipped with SBX40 daughter board can be used as a detector and reader. This article uses the open source written by Nikos on GitHub to use USRP as a reader, and based on this, make appropriate modifications based on experimental conditions and design. The SBX-40 daughter board provides MIMO functionality and provides 40MHz bandwidth. The operating frequency is 400MHz Up to 4400MHz.

        The platform used in the experiment is 64-bit Ubuntu 14.04 and GNU Radio 3.7.4. The selected CRFID tag is the WISP4.1 version with a dipole antenna, and its microcontroller is MSP430F2132. The commercial reader model used is ImpinJ Speedway R420, which is connected to the Laird circularly polarized directional antenna S9028PCL. It can connect up to 4 antennas at the same time, and the antenna gain is 9dBi.

        This paper sets up experimental points in the office, placing sensor nodes in different locations. A stable environment is an office with no people moving around, and an unstable environment is an office with people moving around. It is worth noting that the positions and number of sensor nodes are the same in stable and unstable environments.

        As explained in Section 6, 6.1 illustrates the prediction effect, and 6.2 illustrates the real-time nature of the prediction, which can meet the frequency hopping interval. As shown in Figure 11 in 6.3, our framework has a 20% improvement in throughput compared to previous work in a stable environment. As shown in Figure 12, in an unstable environment, the experimental results fully show the superiority of our framework, which has an 80% -90% improvement in previous work throughput.

Reviewer 4 Report

1. The authors should compare the proposed scheme with "Zeng Y, Nie L. Poster: Channel Prediction Based on BP Neural Network for Backscatter Communication Networks[C]//Proceedings of the 2019 International Conference on Embedded Wireless Systems and Networks. Junction Publishing, 2019: 248-249."

2. The authors should cite their own poster.

3.The complexity of proposed scheme need to be analyzed. Moreover, the proposed scheme will be implemented in which or all sensor nodes should be clarified.

Author Response

Point 1: The authors should compare the proposed scheme with "Zeng Y, Nie L. Poster: Channel Prediction Based on BP Neural Network for Backscatter Communication Networks[C]//Proceedings of the 2019 International Conference on Embedded Wireless Systems and Networks. Junction Publishing, 2019: 248-249."

Response 1: Thanks for your opinion very much! First of all, the special issue we submitted is a journal recommended by the EWSN 2019 conference, which contains articles from Posters and Demos. Secondly, the articles included in the conference are our own articles, but the authors are wrong. We were previously contacted to make changes, but unfortunately the ACM Digital Library has not corrected our author information. Finally, we attach a screenshot, which is a screenshot of the papers included in the EWSN 2019 conference. At the same time, you can log in to the following website for query(http://ewsn2019.thss.tsinghua.edu.cn/call-for-posters.html).

Point 2: The authors should cite their own poster.

Response 2: Thanks very much for your valuable comments. Because the authors of our poster have errors and we are still in the communication stage, we have no way to cite our own poster at this time.

Point 3: The complexity of proposed scheme need to be analyzed. Moreover, the proposed scheme will be implemented in which or all sensor nodes should be clarified.       

Response 3: Thanks very much for your valuable comments.

1)The overall framework performance we measure is mainly reflected by the throughput. As pointed out in 6.3, compared to previous work, our framework achieves an improvement in throughput. The framework is mainly divided into two parts, one is the monitoring module and the other is the prediction module. The data of the monitoring module can be obtained almost in time, and the complexity of the prediction module is mainly reflected by the real-time prediction, that is, the prediction time. As pointed out in 6.2, no matter in a stable environment or an unstable environment, our real-time performance can be controlled within 2s. The prediction time can meet the frequency hopping interval, that is, the switching channel time.

2) The sensor node we use is WISP (Wireless Identification and Sensing Platform) with an acceleration sensor.

We set up experimental points in the office, placing sensor nodes in different locations. A stable environment is an office with no people moving around, and an unstable environment is an office with people moving around. It is worth noting that the positions and number of sensor nodes are the same in stable and unstable environments.

Round 2

Reviewer 3 Report

Your answers to my remarks are not satisfactory.

I have not found any experimental data in Figures 1-2 and 7-12 coherent to your paper's scope. Because you mentioned channel estimation and prediction, it is compulsory to provide testing scenarios in which you have tested the channel information. Because you mentioned RSSI, BER, and SNR I would have expected to find these parameters in your figures. Regarding your statements on stable and unstable environments, people walking by is attributed to attenuation. When you mention an unstable environment you should also mention interference. Its impact is no where fond in your experimental data. You cannot make channel estimation if you do not provide the characteristics of the interference sources.

I did't find any relation between the acceleration that you have measured and the distance. You should provide this relation.

Reviewer 4 Report

No further comments.

Author Response

Thank you for reviewing our article.

Round 3

Reviewer 3 Report

The title of this paper does not reflect the content and issues covered within.

There are no explanations or implementation of a BP neural network (Figure 4 is not related to the article content) Channel prediction is not explained nor implemented in the article. There is a general confusion regarding channel propagation and channel estimation methods. The definition of stable and unstable environment has no relation with the physical phenomena of propagation (reflections, interferences, attenuations). There is no explanations about the spectrum information (how the channels were selected and no experimental data provided: Figures 10 and 11 are not relevant in this sense) Regarding Backscatter Communication Networks, as mentioned in the text of the article: “the environment of the backscatter communication networks is usually complicated, and there are some interferences such as people walking and other wireless networks”. The phenomenon of interference was not tackled in the article, only attenuation caused by people walking by, although there are also types of attenuation that were not considered. In this sense, due to the large variation of the link quality (e.g. RSSI, see Figure 3), BLINK does not offer relevant information because: “the precondition for BLINK [3] is to assume that all nodes experience the same channel quality”. CARA, which: “proposed a channel-aware rate adaptation method considering spatial diversity and frequency diversity, but requires a small interval of time to probe channel quality and increases the overhead of channel probing”actually takes into consideration channel (spectrum) information, something that the proposed methodology of this paper does not provide, only attenuation caused by people walking by.

Related to the previous review round, unfortunately, your answers are not satisfactory:

Point 1: Regarding the following remark: RSSI, BER, SNR (distance), packet loss, and burstiness metric are absent from your measurements.

You have given some explanations in the text, but there is still a lack of correlations between these metrics and your measurements. Only RSSI and packet losses are given in Figure 3 but only three measurements are not enough to evaluate the link quality.

Point 2: There is no coherent answer to this remark:

“There is no information regarding the spectrum: how it was measured and how do you select the channels based on your methodology (See Figure 3). There is no explanation correlated with measurements”.

You have given some explanations regarding how RSSI and packet loss are measured. But how they are used in order to select the channel and use the channel information in correlation with a BP neural network is not presented. In Figure 4 the BPNN is only suggested but not implemented.

Point 3:  “Regarding acceleration, you have presented the sensor's output dependence. If the modulus of acceleration is your chosen norm, then it is not relevant for measuring the distance. You have established a threshold of 5.1, but there is no explanation.” The relation between acceleration and distance is provided by kinematics. This understanding of the basic mechanic principle is absent from your explanation, and should be taking into consideration when establishing a threshold.

Point 4: “Regarding the unstable and stable environments, there is no information (raw data) about their characteristics”.

Figures 7 and 8 do not provide any raw data regarding the characteristics of propagation of the so called ‘stable’ and ‘unstable’ environments.

From your definition it results that: “A stable environment is an office with no people moving around, and an unstable environment is an office with people moving around”. I have a problem with your definition. Your definition is based on what? You have not provided any references in this sense. If it is something you have defined on your own, then you should come with some explanations, and also references in order to justify your definition. There are many books and studies on channel propagation. I did not see any references regarding channel propagation, especially in your ISM band (920-925 MHz). You should understand that the human body does not attenuate the signal too much. Consult this link: https://www.dataloggerinc.com/wp-content/uploads/2016/11/16_Basics_of_signal_attenuation.pdf

I can think of many other sources of attenuation that can attenuate the signal even more than people walking by, for example a large door or wall. I cannot see how this type of attenuation can lead to instability. From Figure 3, one could see that the packet loss rate is between 72% and 99% in the ‘unstable’ environment. What about the stable environment? Do you have a 0% packet loss in the stable environment? Only if you have made the same measurements in an anechoic chamber you could compare results between a stable and an ‘unstable’ environment. Also, you did not mention anywhere the details of your experimental settlement scenarios, the distance between transmitters, how many people are walking by, for what period, are there are other sources of attenuation/interference present in your scenarios.

You have mentioned at the beginning: “the environment of the backscatter communication networks is usually complicated, and there are some interferences such as people walking and other wireless networks” First, people walking by is not a source of interference. Interference can be caused by external signals that operate in the same frequency band as your signal. In your paper you have only tackled the attenuation caused by people walking by, although there also other types of attenuation, like attenuation with distance or caused by other obstacles. Since you have failed to mention what type of interference you have applied to your signal, please remove the word interference from your article. Consult the following link, regarding LoRa interference on RFID in the same frequency band as yours: https://www.ofca.gov.hk/filemanager/ofca/en/content_669/tr201706_01.pdf

If Figure 3 contains only 3 measurements from 3 tags (one from each tag), then it is not sufficient to predict the channel quality accurately. You have mentioned that: “The prediction algorithm predicts the RSSI, packet loss rate, and reading rate of all channels at the next moment”. There is no evidence in your article that you have developed such algorithm for testing and predicting RSSI and packet loss rate. If Figure 3 contains the only proof, then you should stick only to reading rate and throughput analysis and prediction and forget about channel propagation issues (analysis and prediction) which you did not define and test accordingly. Or maybe you did it in Figure 2, but since there is no way to compare it to an unstable environment in terms of channel information and link quality, and thus no one can understand how the Framework overview in Figure 4 really works.

Point 5:  My remark that: “The experimental settlement is not detailed accordingly. The measurements should be clearly presented in order to highlight the selected methodology and the obtained results” is still missing a pertinent answer that should be clearly by Figures 14-15. You said in the article that: “When there are 5 tags, the throughput of our method is 0.5x and 1.4x better than BLINK and CARA” but in your answer to this remark you state that actually your method “has a 20% improvement in throughput compared to previous work in a stable environment. As shown in Figure 12, in an unstable environment, the experimental results fully show the superiority of our framework, which has an 80% -90% improvement in previous work throughput.” The correlation between these statements has serious flaws. As mentioned previously the experimental settlement does not reflect the selected methodology, especially related to BP neural network and channel estimation.
